# Na^+^/H^+^ Exchangers Involve in Regulating the pH-Sensitive Ion Channels in Mouse Sperm

**DOI:** 10.3390/ijms22041612

**Published:** 2021-02-05

**Authors:** Hang Kang, Min Liu, Wei Zhang, Rong-Zu Huang, Na Zhao, Chen Chen, Xu-Hui Zeng

**Affiliations:** 1Institute of Life Science and School of Life Science, Nanchang University, Nanchang 330031, Jiangxi, China; kanghang1991@163.com (H.K.); lmceed@163.com (M.L.); zhangwei332020@163.com (W.Z.); na.zhao@rongdasoft.com (N.Z.); 2Institute of Reproductive Medicine, School of Medicine, Nantong University, Nantong 226019, Jiangsu, China; hrz15880432458@163.com (R.-Z.H.); ntuchen01@163.com (C.C.)

**Keywords:** Na^+^/H^+^ exchangers, KSper, CatSper, cytosolic alkalization, membrane potential, sperm motility

## Abstract

Sperm-specific K^+^ ion channel (KSper) and Ca^2+^ ion channel (CatSper), whose elimination causes male infertility in mice, determine the membrane potential and Ca^2+^ influx, respectively. KSper and CatSper can be activated by cytosolic alkalization, which occurs during sperm going through the alkaline environment of the female reproductive tract. However, which intracellular pH (pH_i_) regulator functionally couples to the activation of KSper/CatSper remains obscure. Although Na^+^/H^+^ exchangers (NHEs) have been implicated to mediate pH_i_ in sperm, there is a lack of direct evidence confirming the functional coupling between NHEs and KSper/CatSper. Here, 5-(*N*,*N*-dimethyl)-amiloride (DMA), an NHEs inhibitor that firstly proved not to affect KSper/CatSper directly, was chosen to examine NHEs function on KSper/CatSper in mouse sperm. The results of patch clamping recordings showed that, when extracellular pH was at the physiological level of 7.4, DMA application caused KSper inhibition and the depolarization of membrane potential when pipette solutions were not pH-buffered. In contrast, these effects were minimized when pipette solutions were pH-buffered, indicating that they solely resulted from pH_i_ acidification caused by NHEs inhibition. Similarly, DMA treatment reduced CatSper current and intracellular Ca^2+^, effects also dependent on the buffer capacity of pH in pipette solutions. The impairment of sperm motility was also observed after DMA incubation. These results manifested that NHEs activity is coupled to the activation of KSper/CatSper under physiological conditions.

## 1. Introduction

Ion channels play critical roles in the regulation of cellular physiological activity. In mammalian sperm, it has been found that sperm-specific K^+^ ion channel (KSper) and Ca^2+^ ion channel (CatSper), as two kinds of vital ion channels, were responsible for the modulation of membrane potential and Ca^2+^ influx, respectively [1,2,3,4,5,6,7,8]. Benefiting the established knockout models, the elimination of KSper or CatSper channels in mice caused male infertility, which resulted from the severe impairments of sperm motility and functions [4,9,10,11,12,13,14,15]. More importantly, the patients who exhibited the abnormal regulation of membrane potential primarily mediated by KSper or the mutation of CatSper-related gene also might suffer from the infertile syndrome [16,17,18]. As the indispensability of KSper and CatSper channels in the fertilizing capacity of sperm, fully illuminating the physiological regulation of these two ion channels is, therefore, worth carrying out.

With the success of the sperm patch-clamp technique, it has been well known that pH_i_ was the dominant inducer for the functional activation of KSper and CatSper channels in mouse sperm [1,5,9,10,15,19]. During the fertilization process, the elevation of pH_i_ could hyperpolarize KSper-mediated membrane potential and promote CatSper-mediated Ca^2+^ influx [1,10,19,20]. For the KSper channel, Slo3, as the main subunit which accounted for KSper conductance, was found to be regulated by pH in an allosteric manner [21,22]. Deletion of Slo3 in mouse sperm also resulted in the failure of alkalization-induced hyperpolarization [10]. For the CatSper channel, early reports have inferred that the property of pH-sensitivity in the CatSper channel was presumably due to the highly enriched histidine in CatSper1 protein [4]. However, a recent study found that EF-hand calcium-binding domain-containing protein 9 (EFCAB9), which was associated with CatSperζ, was of significance for the pH-sensitive activation of CatSper. Alkalized pH_i_ enhanced the Ca^2+^ sensitivity of EFCAB9 and diminished the interaction of EFCAB9 and CatSperζ, both of which triggered the activation of CatSper [15]. Thus, the acquisition of cytoplasmic alkalization is perceived as a pivotal factor that facilitates the opening probability of KSper and CatSper.

Several ion channels and transporters have been considered as the underlying pH_i_ regulators of sperm [23]. Among them, Na^+^/H^+^ exchangers (NHEs), as the main membrane H^+^ transporters, were suggested to mediate the proton efflux via responding to the extracellular ascending pH in the female reproductive tract [23,24]. However, whether NHEs activity was functionally coupled to the activation of KSper and CatSper was not yet elucidated. In this study, to define the function of NHEs on the physiological activation of KSper and CatSper in epididymal sperm of mice, we utilized 5-(*N*,*N*-dimethyl)-amiloride (also named DMA), a potent inhibitor of sperm NHEs which does not affect KSper and CatSper channel directly. We demonstrated by patch-clamp recordings, Ca^2+^ fluorimetry, and motility measurements that the treatment of DMA remarkably attenuated the potentiation of KSper and CatSper channels through impairing the regulation of pH_i_, and thereby, reduced intracellular Ca^2+^ concentration and motility of sperm. This study may shed insight into the coordinated interaction between key components during sperm function regulation.

## 2. Results

### 2.1. DMA Impaired the Activation of KSper Channel via Inhibiting NHEs Function

Although DMA was reported as the potent inhibitor for NHEs (IC_50_ = 6.9 μM) [25], whether the physiological activity of NHEs in mouse sperm was also impaired by the application of DMA needs to be verified. By employing the pH indicator BCECF, it was found that 20 μM DMA acidified the cytoplasm of mouse sperm (Appendix A), which indicated the involvement of DMA-sensitive NHEs in the regulation of pH_i_. Ammonium chloride (NH_4_Cl), a definite stimulus of intracellular alkalization for sperm, was employed as a positive control (Appendix A). Therefore, DMA was utilized to evaluate the effect of NHEs activity on the pH-dependent KSper and CatSper channels in mouse sperm. 

The coupling between NHEs and KSper was examined first. To exclude the possibility that DMA might affect KSper directly, the effect of DMA on KSper currents was tested under pH-buffered pipette solution, and it showed that DMA had no effect on KSper currents (Appendix A), indicating that DMA is suitable to elucidate the coupling of NHEs activity on the activation of KSper channel. The pipette pH was set to 7.0, a level at which both NHEs and KSper should be activated [1,26]. The results showed that DMA only brought about a slight reduction on KSper currents in the pipette solution of pH 7.0 containing the pH buffer (Figure 1a,c). In contrast, when pH buffer in the pipette solution was removed to allow the change of pH in the sperm cytoplasm, DMA inhibited the KSper currents significantly (Figure 1b,c). Since the only difference between Figure 1a and 1b was whether the pipette pH was buffered, the inhibition of KSper currents recorded in Figure 1b should only result from the intracellular acidification caused by the inhibition of DMA on NHEs, thus supporting the coupling between NHEs and KSper.

### 2.2. NHEs Inhibition Resulted in the Depolarization of Membrane Potential

The hyperpolarization of membrane potential during the capacitation process is necessary for the fertilizing capacity of sperm [10,16,19,27,28]. Given that KSper dominates the maintenance of membrane potential in mouse sperm [1], the transfer of DMA inhibition on NHEs to the reduction of KSper activity should be reflected as the change of sperm membrane potential. Since DMA may inhibit other channels/transporters such as the epithelial Na^+^ channel [25,29], the effect of DMA on the membrane potential with a pH-buffered pipette solution was examined first. Current clamp recordings showed that the membrane potential of mouse sperm had no significant alteration after adding 20 μM DMA at various pH_i_ (Figure 2), suggesting that pH-independent hyperpolarization of membrane potential caused by DMA can be ignored. 

Subsequently, we removed the pH buffer in the pipette solutions and explored whether NHEs inhibition caused by DMA could depolarize sperm membrane potential. After adding 20 μM DMA, the membrane potential of mouse sperm depolarized significantly at pH_i_ 6.0 (−1.00 ± 2.58 mV to 20.31 ± 4.32 mV; *p* < 0.001), 7.0 (−29.09 ± 2.33 mV to −19.54 ± 3.08 mV; *p* < 0.05) or 8.0 (−40.47 ± 1.56 mV to −28.60 ± 1.78 mV; *p* < 0.001), respectively (Figure 3). Of note, the depolarized level at pH_i_ 7.0 or 8.0 was less dramatic than that at pH_i_ 6.0, possibly due to the compromised activity of NHEs at pH_i_ 7.0 and 8.0. Consistently, DMA could partially reduce the hyperpolarization of membrane potential caused by 10 mM NH_4_Cl, which is usually used to alkalize cells at low pH_i_ (Appendix A). Thus, it was concluded that NHEs inhibition remarkably suppressed the activation of the pH sensitive KSper channel and depolarized the membrane potential in mouse sperm.

### 2.3. Alkalization-Activated CatSper Channel and NHEs Were Functional Coupled

Since CatSper is also pH-dependent, we further investigated whether the inhibition of DMA on NHEs could affect the activity of CatSper. Similar to the strategy of the KSper study, the effect of DMA on CatSper currents was compared by using pipette solutions with or without a pH buffer. When the pipette solution was pH-buffered, the inward current of the CatSper channel was not impaired in the presence of DMA (Figure 4a,c). In contrast, DMA dramatically diminished the CatSper current when the pH buffer was eliminated from the pipette solution (Figure 4b,c). Statistical analysis showed that the inward CatSper currents at −80 mV decreased from −567.1 pA ± 65.15 pA to −234.6 ± 44.29 pA (Figure 4c). It was worth noting that CatSper current at high positive voltage decreased after adding DMA in Figure 4a. This decrease remained even as 50 mM of HEPES was added in the pipette solution to buffer the pH_i_ (Appendix A), implying a disturbance of HEPES on the outward currents through CatSper. Nevertheless, this unphysiological effect should not affect the idea that the activation of the CatSper channel was also attenuated after inhibiting NHEs by DMA.

### 2.4. DMA Treatment Reduced Intracellular Ca^2+^ Concentration and Impaired Sperm Motility

Internal Ca^2+^ concentration regulates almost all of the physiological activities of sperm [30]. Considering that CatSper is the unique approach for Ca^2+^ influx and the hyperpolarization of membrane potential modulated by KSper provided a driving force of Ca^2+^, we speculated that the homeostasis of cytoplasmic Ca^2+^ would be disturbed by DMA. Indeed, the treatment of DMA pronouncedly down-regulated Ca^2+^ signals (Figure 5a,b). Furthermore, the results showed that the total motility and progressive motility of mouse sperm incubated with DMA exhibited a severe reduction (Figure 5c,d). Surprisingly, the velocity parameters, including those reflecting sperm hyperactivation which is dependent on CatSper, were not affected by DMA (Appendix A). Although the underlying mechanism to explain this apparently inconsistent information needs further investigation, the results generally support that the activity of NHEs is functionally coupled to the activation of CatSper under physiological conditions.

## 3. Discussion

The pH-regulated activation of KSper and CatSper is important for almost all functional aspects of sperm, such as capacitation, hyperactivation, acrosome reaction, and sperm-egg fusion [23]. By utilizing patch clamping recordings in single sperm together with an NHEs inhibitor DMA which was proved not to affect KSper and CatSper directly, this study clearly showed that the inhibition of NHEs decreased the activities of both KSper and CatSper, supporting the idea that NHEs activity is functionally coupled to the activation of KSper and CatSper channels. In another word, under physiological conditions, NHEs-mediated cytoplasmic alkalization should potentiate the pH-sensitive KSper and CatSper, and as a consequence, regulate sperm functions. 

As the main transporters for eliminating the cytosolic protons, six isoforms of NHEs: NHE1 [31,32,33], NHE5 [34], NHE8 [35], sNHE [36,37], NHA1 [38], and NHA2 [39] had been reported to be expressed in sperm. However, which NHE is the determinant modulator for the activation of KSper and CatSper in murine sperm remained ambiguous. Considering that NHE1 and NHE2 were the primary targets for DMA [40], we adopted another selective NHE1 antagonist cariporide to verify NHE1 contribution in the activation of CatSper [32,40]. However, 10 μM cariporide failed to diminish the CatSper current in the pipette solution without a pH buffer (Appendix A). Previous reports manifested that NHE1 was expressed in the midpiece of sperm [34], while this location could not detect any appreciable currents of KSper or CatSper [1,5]. Therefore, we speculated that NHE1 activity did not account for the activation of pH-sensitive ion channels.

sNHE and NHAs expressed in the principal piece of sperm were also candidates for mediating the activation of pH-dependent ion channels [36,39]. Given that NHAs were insensitive to amiloride [41], we supposed that KSper and CatSper were functionally coupled to sNHE. Unfortunately, there is no sNHE specific blocker or commercially available antibodies of sNHE which were capable of inhibiting CatSper currents (Appendix A). Thus, we could not clarify the contribution of sNHE on the activation of the CatSper channel. Nevertheless, several studies implied that sNHE worked hand-in-hand with the KSper channel owing to the occurrence of the voltage sensor in sNHE and the pH sensitivity in KSper [9,20], supporting a functional interaction between sNHE and KSper. In terms of the CatSper channel, however, there was no measurable difference of CatSper currents recorded from WT and *sNHE^−/−^* sperm [12]. In that study, the pipette solution contained 10 mM of HEPES, which would minimize the pH regulation effects of sNHE on CatSper [5,12]. Comparison of KSper and CatSper currents recorded with pH-unbuffered pipette solution from WT and *sNHE^−/−^* sperm should clarify the contribution of sNHE on the activation of KSper and CatSper channels.

This study revealed a functional coupling between NHEs and pH-dependent KSper and CatSper channels in mouse sperm. Is a similar mechanism shared in human sperm? Human KSper exhibits less pH sensitivity than mouse KSper [2,3]. Besides, the regulatory mechanism of pH_i_ may be different between human and mouse sperm [42]. Although it has been reported that NHE1 was expressed in human spermatozoa, a broad-spectrum NHE inhibitor EIPA could not block the initiation of acrosome reaction induced by progesterone [32]. Moreover, the voltage-gated proton channel Hv1 has been proposed as the major regulator for acid extrusion and physiological functions of human sperm [43]. Therefore, whether the NHEs-mediated cytoplasmic alkalization contributes to the potentiation of KSper and CatSper activities in human sperm needs further investigation. 

## 4. Materials and Methods

### 4.1. Reagents

Fluo 4-AM and BCECF-AM were purchased from Molecular Probes (Thermo Fisher, Waltham, MA, USA). Bovine serum albumin (BSA) was purchased from Sangon Biotech (Shanghai, China). The anti-sNHE antibodies were purchased from Abcam (Cambridge, MA, USA) and Invitrogen (Thermo Fisher, Waltham, MA, USA). Other chemicals were obtained from Sigma-Aldrich (St Louis, MO, USA).

### 4.2. Animal Treatment and Sperm Preparation

The C57BL/6 mice (25–35 g; 12–20 weeks) for experiments were received from the Animal Center of Nanchang University. All animal procedures were performed in accord with the recommendations of the Animal Center of Nanchang University guidelines and approved by the Animal Research Ethics Committee of Nanchang University (SYXK2015-0002, January 2015, Nanchang, China). Mice were treated humanely and housed at the temperature of 20–25 °C.

Sperm were obtained from the isolated epididymides of male mice. For measurements of intracellular pH, Ca^2+^, and motility, the caudal epididymis was gently perforated to collect mature sperm. For patch-clamp recordings, the corpus epididymis was snipped into several parts to collect sperm containing the robust cytoplasmic droplet required for gigaohm seal. Although sperm in corpus epididymis are less mature than sperm in caudal epididymis [44], it was suggested that there are no significant differences in the properties of ion channels between the corpus and caudal epididymis of sperm [45]. Afterward, Minced mouse epididymis was rinsed in HS solution (135 mM NaCl, 5 mM KCl, 1 mM MgSO_4_, 2 mM CaCl_2_, 20 mM Hepes, 5 mM Glucose, 10 mM Lactic acid, and 1 mM Na-pyruvate at pH 7.4 with NaOH). Sperm was shaken slowly for about 20 min to swim out at 37 °C. Then, the supernatant was transferred into a 1.5 ml centrifuge tube for experiments.

### 4.3. Electrophysiology

Whole-cell recordings were conducted on non-capacitated sperm as reported [19]. Briefly, mouse sperm in corpus epididymis were isolated into HS solution. Gigaohm seals (>3 GΩ) between patch pipette (15–25 MΩ) and sperm cytosolic droplet were gained. Transitions into the whole-cell mode were made by applying 5 ms, 350–500 mV voltage pulses, and combined with light suction. For voltage-clamp recordings of KSper, the extracellular K^+^ symmetrical solution contained 160 mM KOH, 10 mM Hepes, 150 mM methanesulfonic acid (Mes), 2 mM Ca(OH)_2_, adjusted to pH 7.4 with KOH. A strong pH-buffered pipette solution contained 155 mM KOH, 5 mM KCl, 10 mM 1,2-bis (2-aminoph enoxy) ethane-*N*,*N*,*N*′,*N*′-tetraacetic acid (BAPTA), 20 mM Hepes, and 115 mM Mes adjusted to pH 8.0 with KOH. The pipette solution without pH buffer or with pH buffer contained: 140 mM KOH, 15 mM NaCl, 5 mM KCl, 1 mM BAPTA, 3 mM MgATP, 0.5 mM Na_3_GTP, 0 or 20 mM HEPES, adjusted to pH 7.0 with KOH. For current-clamp recordings of KSper, HS was used for primary extracellular solution for the current clamp. The pipette solution without pH buffer or with pH buffer contained 144 mM KOH, 5 mM KCl, 10 mM NaCl, 3 mM MgATP, 0.5 mM Na_3_GTP, 1 mM BAPTA, 140 mM Mes, and 0 or 20 mM Hepes with pH 6.0, 7.0, and 8.0 adjusted with Mes or KOH. For whole-cell recordings of mouse monovalent CatSper currents, the pipette solution without pH buffer or with pH buffer was filled with: 135 mM Cs-Mes, 5 mM CsCl, 0 or 20 mM HEPES, 10 mM EGTA adjusted to pH 7.2 with CsOH. The sodium-based divalent solution (DVF) comprised 150 mM sodium gluconate, 20 mM HEPES and 5 mM Na_3_HEDTA adjusted to pH 7.4 with NaOH. The extracellular solutions were applied directly via a local perfusion system. All currents and membrane potentials were recorded by utilizing Axon 200B amplifier (Molecular Devices, San Jose, CA, USA). Signals were filtered at 1 kHz and sampled at 10 kHz. All experiments were at room temperature (~22–25 °C). Data were analyzed with Clampfit 10.4 (Molecular Devices, San Jose, CA, USA) and Grapher 8 (Golden Software Inc., Golden, CO, USA).

### 4.4. Determination of Sperm Intracellular pH and Ca^2+^

Sperm intracellular pH and Ca^2+^ were examined by fluorescence recordings as previously reported [8]. Sperm obtained from the caudal epididymis were loaded with 0.5 μM pH-sensitive dye BCECF-AM or 5 μM Ca^2+^ indicator Fluo-4 AM in the presence of F-127 (0.1% v/v) for 30 min at 37 °C. Then, sperm were washed once at 300 g for 5 min and resuspended in HS solution. For the recordings, these sperm were loaded in the multimode plate reader (PerkinElmer, Waltham, MA, USA) at 30 °C. When the recordings were stable, HS or 20 μM DMA was added. 10 mM NH_4_Cl was a positive control. The relative changes of intracellular pH and Ca^2+^ were showed by (ΔF)/F0 × 100%. ΔF indicates the fluorescence changes before and after the injection of solutions. F0 represents the stable value of basal fluorescence.

### 4.5. Assessment of Sperm Motility

Mouse sperm for the detection of motility were collected from the caudal epididymis. HS solution containing 25 mM NaHCO_3_ and 5 mg/mL BSA was prepared for measuring sperm motility. Two sides of caudal epididymides in one mouse were divided into the control group and the experimental group (20 μM DMA), respectively. Caudal epididymis was gently perforated to allow sperm to swim-out for 15 min at 37 °C. Then, sperm samples were injected into 80 μm deep chambers (Hamilton-Throne, Beverly, MA, USA). Sperm motility was analyzed using a computer-assisted sperm analysis (CASA) system (Hamilton-Throne, Beverly, MA, USA) on a Zeiss microscope with a 10 × objective (Oberkochen, German) every 30 min. More than 200 motile sperm were examined. The parameters including the percentage of motile and progressive motile of sperm, average path velocity (VAP), curvilinear velocity (VCL), straight-line velocity (VSL), amplitude of lateral head displacement (ALH), beat cross frequency (BCF) and linearity (LIN) were calculated.

### 4.6. Statistical Analysis

Data were expressed as mean ± SEM. Statistical significance in the results of sperm total motility and progressive motility were determined by employing an unpaired *t*-test. Other results were assessed by a paired *t*-test (for two group comparisons) and a one-way ANOVA (for multi-group comparisons). A statistically significant difference was set at *p* < 0.05. All statistical analyses were performed in GraphPad Prism, version 5.0 software (San Diego, CA, USA).

## Figures and Tables

**Figure 1 ijms-22-01612-f001:**
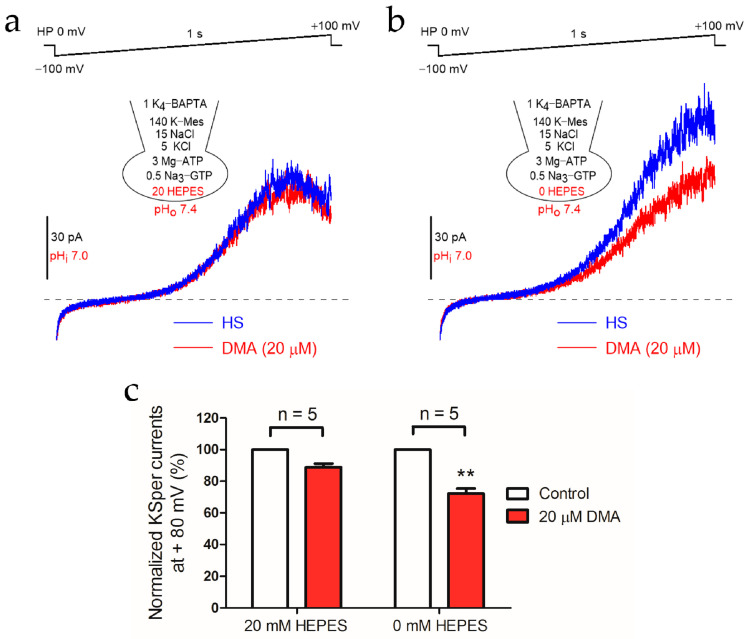
5-(*N*,*N*-dimethyl)-amiloride (DMA) prominently reduced K^+^ ion channel (KSper) currents after the removal of pH buffer in the pipette solution. Representative patch-clamp recordings of KSper currents in the absence or presence of 20 μM DMA elicited by 1 s voltage ramp from −100 mV to + 100 mV with a 20 mM 4-(2-hydroxyethyl)-1-piperazineethanesulfonic acid (HEPES) (**a**) or 0 mM HEPES (**b**) pipette solution at pH_i_ 7.0. The high saline solution (HS) containing 135 mM Na^+^ and 2 mM Ca^2+^ was employed to maintain Na^+^/H^+^ exchangers (NHEs) activity and block the CatSper current, respectively. (**c**) Statistical analysis of the mean normalized KSper currents at + 80 mV recorded in the pipette solution containing 20 mM HEPES (*n* = 5) or 0 mM HEPES (*n* = 5). Data are expressed as mean ± standard error of the mean (SEM). ** *p* < 0.01 (paired *t*-test).

**Figure 2 ijms-22-01612-f002:**
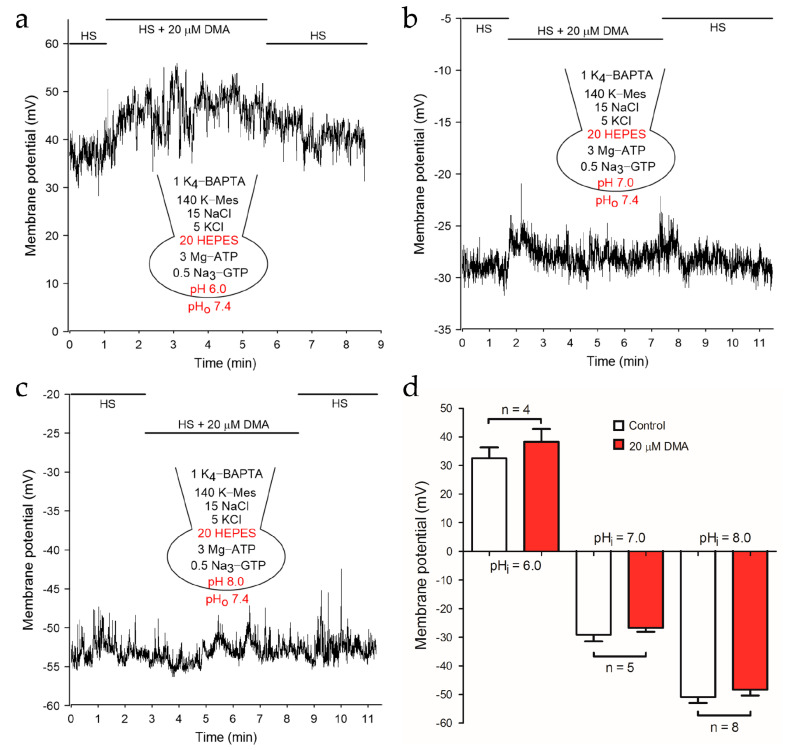
DMA failed to depolarize the membrane potential in a pH buffered pipette solution. Representative traces of membrane potentials recorded by the current clamp at pH_i_ 6.0 (**a**), pH_i_ 7.0 (**b**), pH_i_ 8.0 (**c**) was shown in the absence or presence of 20 μM DMA. HS solution containing 135 mM Na^+^ was employed to maintain NHEs activity. (**d**) Mean amplitudes of membrane potential before and after the treatment of DMA at pH_i_ 6.0 (*n* = 4), pH_i_ 7.0 (*n* = 5) and pH_i_ 8.0 (*n* = 8). Data are expressed as mean ± SEM.

**Figure 3 ijms-22-01612-f003:**
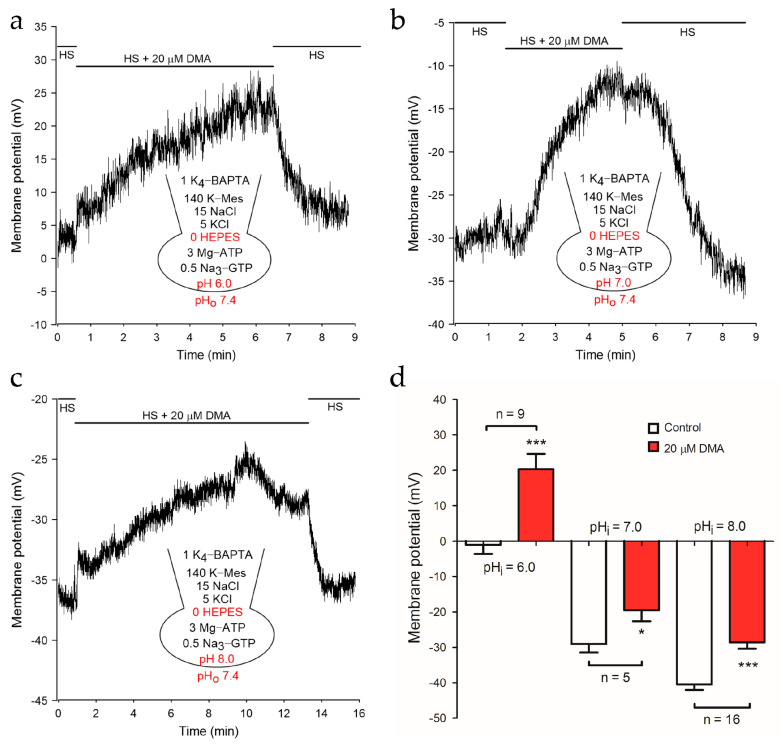
The membrane potential of mouse spermatozoa was depolarized by DMA in the pipette solution with no pH buffer. Sperm membrane potential was reversibly depolarized by 20 μM DMA at pH_i_ 6.0 (**a**), pH_i_ 7.0 (**b**), pH_i_ 8.0 (**c**). Representatives of three current-clamp recordings are shown respectively. HS solution containing 135 mM Na^+^ was employed to maintain NHEs activity. (**d**) Mean amplitudes of membrane potential before and after the treatment of DMA at pH_i_ 6.0 (*n* = 9), pH_i_ 7.0 (*n* = 5) and pH_i_ 8.0 (*n* = 16). Data are expressed as mean ± SEM. * *p* < 0.05, *** *p* < 0.001 (paired *t*-test).

**Figure 4 ijms-22-01612-f004:**
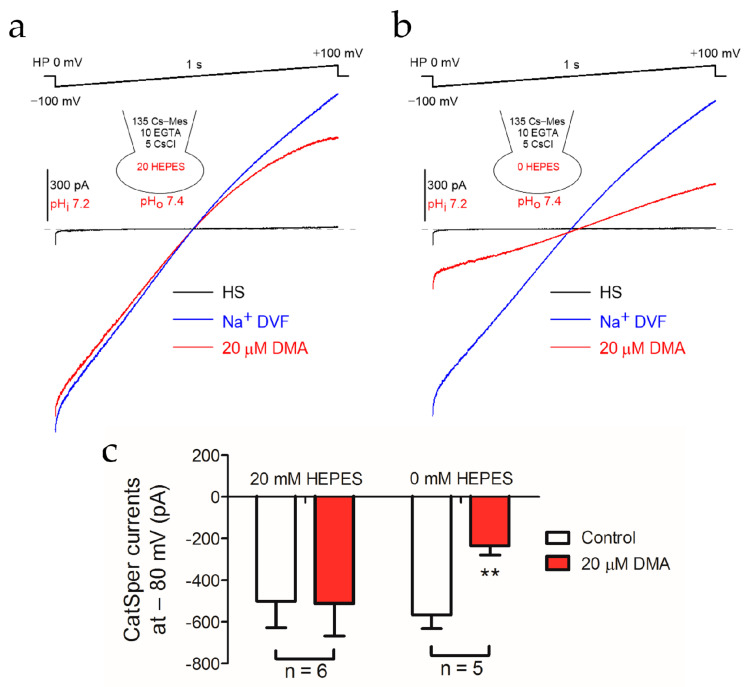
DMA prominently inhibited the Ca^2+^ ion channel (CatSper) currents via the acidification of pH_i_. Representative patch-clamp recordings of CatSper currents in the absence or presence of 20 μM DMA elicited by 1 s voltage ramp from −100 mV to +100 mV with a 20 mM HEPES (**a**) or 0 mM HEPES (**b**) pipette solution at pH_i_ 7.2. The sodium-based divalent free solution (Na^+^ DVF) was employed to record the monovalent CatSper current and maintain NHEs activity. (**c**) Statistical analysis of the mean CatSper currents at −80 mV recorded in the pipette solution containing 20 mM HEPES (*n* = 6) or 0 mM HEPES (*n* = 5). Data are expressed as mean ± SEM. ** *p* < 0.01 (paired *t*-test).

**Figure 5 ijms-22-01612-f005:**
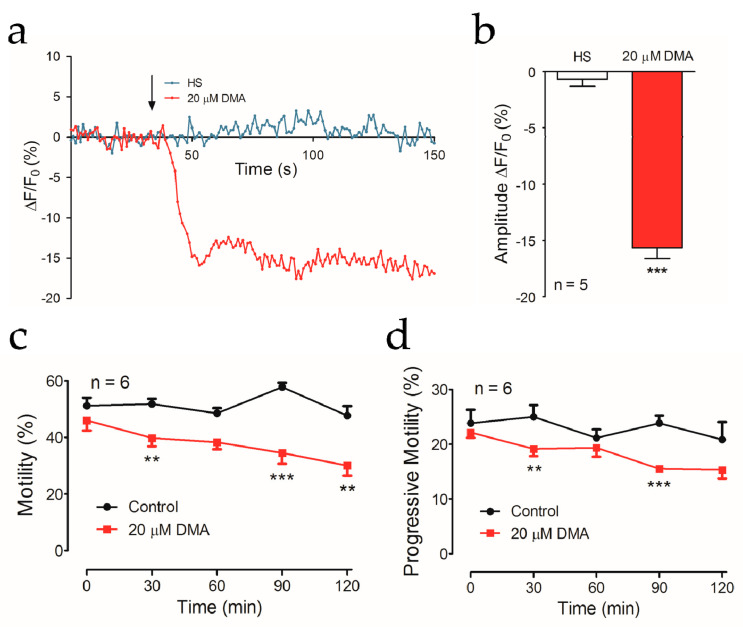
The effect of NHEs inhibition by DMA on intracellular Ca^2+^ and motility of mouse sperm were evaluated. (**a**) The fluorescence trace of intracellular Ca^2+^ signals was shown. Arrow indicates the time point of HS solution or DMA injection. (**b**) Mean fluorescence intensities of Ca^2+^ signals in the absence and presence of DMA. *n* = 5 (*** *p* < 0.001, paired *t*-test). The effect of DMA on the total motility (**c**) or the progressive motility (**d**) of mouse sperm at 0, 30, 60, 90, and 120 min was shown. Data are expressed as mean ± SEM. *n* = 6 (** *p* < 0.01; *** *p* < 0.001, unpaired *t*-test).

## Data Availability

Data is contained within the Appendix A.

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
