# Peer review of "Na+/H+ Exchangers Involve in Regulating the pH-Sensitive Ion Channels in Mouse Sperm"

_ijms, 2021, doi:10.3390/ijms22041612_

Round 1

Reviewer 1 Report

Dear Authors,

firstly, I would like to apologize for the delay with a review of your manuscript entitled: "Na+/H+ exchangers involve in regulating the pH-sensitive ion channels in mouse sperm".

I have found your study very interesting with a high impact on knowledge of sperm physiology based on the solid method which can further be an inspiration for another scientist in this research field. 

I really appreciate the author's readable writing style. All sections are clearly and aptly written. 

I only have a few comments:

In the Abstract I suggest mentioning methods used in the study.

In the section Material and Methods I have one question.

Why authors did not analyze ALH, BCF, and LIN as ones of important kinetic parameters for sperm capacitation analysis? It will be also very interesting to see changes in that

In section results Fig. 5c and 5d I suggest modifying x-axis to starting at zero

In the Discussion, I have one question in ln 212. Is there any reference as the basis for this exact concentration of HEPES used for this purpose?

Thank you for your work.

I wish you many more successes.

My best regards.

Author Response

Response to reviewer 1

Firstly, I would like to apologize for the delay with a review of your manuscript entitled: "Na+/H+ exchangers involve in regulating the pH-sensitive ion channels in mouse sperm".

I have found your study very interesting with a high impact on knowledge of sperm physiology based on the solid method which can further be an inspiration for another scientist in this research field. 

I really appreciate the author's readable writing style. All sections are clearly and aptly written. 

Response: We really appreciate that you think this study is interesting and valuable to sperm physiology. In fact, it’s regretful that we did not identify which NHEs couple to Ksper/CatSper because the lack of proper pharmacological tools. Thanks for your encouraging comments again.

I only have a few comments:

In the Abstract I suggest mentioning methods used in the study.

Response: Thanks for the suggestion. Because patch clamping was the major method used in this study, in the abstract the sentence “The results showed that, when extracellular pH…” in ln 20 is revised to “The results of patch clamping recordings showed that, when extracellular pH…”.

In the section Material and Methods I have one question.

Why authors did not analyze ALH, BCF, and LIN as ones of important kinetic parameters for sperm capacitation analysis? It will be also very interesting to see changes in that.

Response: We agree that this information should help the readers understand the effect of DMA better. In fact, we did analyze ALH, BCF and LIN, and we did not show this information only because DMA also did not affect ALH, BCF and LIN. Taking your suggestion, we append this information in the revised Fig. S5.

In section results Fig. 5c and 5d I suggest modifying x-axis to starting at zero

Response: Did you mean to start the y-axis of Fig. 5c and 5d at zero? Since the x-axis of those panels already started at zero in our previous version, now we modify the y-axis starting at zero in this new version. Please see the revised manuscript (ln 175).

In the Discussion, I have one question in ln 212. Is there any reference as the basis for this exact concentration of HEPES used for this purpose?

Response: Thanks for raising this question. To our knowledge, for patch clamp recordings of sperm, 10 mM HEPES was usually added in the pipette solution to keep the pH unchanged (Kirichok Y et al. Nature. 2006; Qi H et al. PNAS. 2007), so we speculate that the effect of NHE on pH regulation was minimal. Accordingly, we cite these two referees in the revised version (ln 215).

Reviewer 2 Report

This is an interesting and comprehensive study on the physiological relationship between NHEs and Ksperm and Catsperm channels in mouse. The research conducted is very interesting and the manuscript of high-quality.

I just have some minor comments to note:

  • At the end of the Introduction authors must indicate the full name of the DMA inhibitor.
  • After revising the Material and Methods section I realised that the study was performed in epididymal spermatozoa. Therefore, authors must clearly indicate either at the end of the Introduction or at the beginning of the Results that they used epididymal sperm.
  • Moreover, for electrophysiologic assay authors collected sperm cells from the epididymal corpus, whereas for measurements of intracellular pH and Ca2+ and sperm motility, they collected sperm cells from the epididymal cauda. Considering that sperm cells from the epididymal corpus are not completely mature, please, provide some references indicating that the electrophysiological studies on corpus sperm can be extrapolated to mature sperm.
  • Please, provide a description of the procedure followed to collect sperm from the epididymal corpus and the epididymal cauda.

Author Response

Response to reviewer 2

This is an interesting and comprehensive study on the physiological relationship between NHEs and Ksperm and Catsperm channels in mouse. The research conducted is very interesting and the manuscript of high-quality.

I just have some minor comments to note:

At the end of the Introduction authors must indicate the full name of the DMA inhibitor.

Response: We really appreciate your opinion that establishing the functional relationship between NHEs and KSper/CatSper is an interesting conduct. Taking your suggestion, the sentence “we utilized a potent inhibitor of sperm NHEs named DMA not affecting KSper and CatSper channel directly” in ln 66 to 68 is revised to “we utilized 5- (N,N-dimethyl)- amiloride (also named DMA), a potent inhibitor of sperm NHEs which does not affect KSper and CatSper channel directly”.

After revising the Material and Methods section I realized that the study was performed in epididymal spermatozoa. Therefore, authors must clearly indicate either at the end of the Introduction or at the beginning of the Results that they used epididymal sperm.

Response: Thank a lot for the suggestion. Accordingly, at the end of Introduction, the sentence in ln 66 is revised to “to define the function of NHEs on the physiological activation of KSper and CatSper in epididymal sperm of mouse”.

Moreover, for electrophysiologic assay authors collected sperm cells from the epididymal corpus, whereas for measurements of intracellular pH and Ca2+ and sperm motility, they collected sperm cells from the epididymal cauda. Considering that sperm cells from the epididymal corpus are not completely mature, please, provide some references indicating that the electrophysiological studies on corpus sperm can be extrapolated to mature sperm.

Response: This a very pertinent comment. Interestingly, almost all of the electrophysiologic recordings were employed on corpus sperm while more matured cauda sperm were collected for functional assay (see Kirichok Y et al. Nature. 2006; Qi H et al. PNAS. 2007; Navarro B et al. PNAS. 2007 and Zeng XH et al. PNAS. 2011). Based on our experience, it might be because that obtaining high resistance sealing from cauda sperm appears much more difficult than from corpus sperm, on which the droplet required for sealing tends to lose. Nevertheless, there was a review sharing the idea that there is no significant differences between ion channels in corpus and caudal mouse spermatozoa (information from Lishko PV et al. Methods Enzymol. 2013), implying that electrophysiological studies on corpus sperm can be extrapolated to mature sperm. We add this information in the methods section (ln 247 to 249).

Please, provide a description of the procedure followed to collect sperm from the epididymal corpus and the epididymal cauda.

Response 4: Thanks for the suggestion. In Materials and Methods, the corresponding sentences are revised as “Sperm were obtained from the isolated epididymides of male mice. For measurements of intracellular pH, Ca2+ and motility, the caudal epididymis was gently perforated to collect mature sperm. For patch clamp recordings, the corpus epididymis was snipped into several parts to collect sperm...” (ln 243 to 247)